# Revisiting the Asymmetric Matching Pennies Contradiction in China

**DOI:** 10.3390/bs13090757

**Published:** 2023-09-12

**Authors:** Ailin Leng, Zeng Lian, Jaimie W. Lien, Jie Zheng

**Affiliations:** 1Center for Economic Research, Shandong University, Jinan 250100, Chinajie.academic@gmail.com (J.Z.); 2International Business School, Beijing Foreign Studies University, Beijing 100089, China

**Keywords:** mixed-strategy equilibria, experiments, asymmetric matching pennies, cultural difference

## Abstract

The asymmetric matching pennies contradiction posits that contrary to the prediction of mixed-strategy Nash equilibrium, experimental subjects’ choices are, in practice, based heavily on the magnitudes of their own payoffs. Own-payoff effects are robustly confirmed in the literature. Closely following the experimental setups in the literature which support the contradiction, we conduct a series of asymmetric matching pennies games in China, hypothesizing play which is closer to equilibrium frequencies than previously found. Contrary to previous experiments which were conducted in the United States, we find that there are essentially no own-payoff effects among Row players who face large payoff asymmetry. In a Quantal Response Equilibrium framework allowing for altruism or spite, the behavior of our subjects corresponded to a positive spite parameter, whereas the results of previous studies corresponded to altruism. Our results may be consistent with recent psychology literature that finds people from collectivist cultures are substantially more adept at taking the perspective of others compared with people from individualist cultures, a feature of the reasoning needed to obtain mixed-strategy equilibrium.

## 1. Introduction

Mixed-strategy equilibria are critical for the existence of Nash equilibrium in games, but they have posed both theoretical and empirical challenges in justifying their use as a solution concept. The classic theoretical concerns address how to justify players’ pure strategy mixtures in equilibrium, when each player mixes so as to make the other player fully indifferent between any mixing over nondominated pure strategies (see Osborne and Rubinstein [1] for a discussion). However, in addition to theoretical issues, there are also challenges in terms of empirical validity.

Goeree and Holt [2] found that for the case of symmetric matching pennies games with unique mixed-strategy equilibrium, the equilibrium prediction holds up fairly well on human subjects. However, in the case of a highly asymmetric payoff version of the game, the equilibrium prediction fails gravely, as players seem to gravitate heavily towards their own highest payoff opportunity rather than randomizing such that their opponent has no room to exploit them in expectation. Their hypothesis is that the concept of mixed-strategy equilibrium seems to only work coincidentally in the case of symmetry, a conjecture which, if shown to be true, implies the mixed-strategy equilibrium concept may not be an appropriate description of human strategic behavior for situations in which there are large payoff asymmetries involved.

Other studies have robustly confirmed that subjects display “own-payoff” effects in the laboratory for the asymmetric matching pennies class of games under a variety of conditions. Ochs [3] studies asymmetric matching pennies games under repetition, finding that even after 50 rounds of repetition, subjects still gravitate toward their own highest payoff significantly more than equilibrium predicts. These effects are confirmed in the experimental results of McKelvey et al. [4] and Goeree et al. [5], who study individual heterogeneity in own-payoff effects and the influence of risk-aversion in these types of games, respectively.

In our design, subjects in China play repeatedly (50 rounds) under either random rematching or fixed partner matching and can observe the history of play between themselves and their partner(s). While this differs from the set up in Goeree and Holt [2], which implemented a one-shot version of the games, Ochs [3], McKelvey et al. [4] and Goeree et al. [5] also implemented designs which repeated play of the games at least 10 times. We find that subjects adhered to the equilibrium prediction in the aggregate more closely compared with previous studies. Specifically, our Row players, whose payoffs are highly asymmetric, tend to play pure strategy frequencies very close to the equilibrium prediction dictated by his or her partner’s payoffs, and they do not show any sign of the “own-payoff” effects prevalent in previous studies. Our Column players, whose equilibrium mixture is very asymmetric, as dictated by the Row players’ payoffs, show evidence of learning dynamics. That is, over time, Column players’ randomization in the aggregate moves significantly in the direction predicted by equilibrium. The difference between our results and the previous results in the literature can also be understood in a Quantal Response Equilibrium framework, which allows for altruism or spite. While the behavior of subjects in Goeree and Holt [2] corresponds to players with altruistic preferences [6], in the current study, players’ behavior corresponds to spiteful preferences.

Henrich et al. [7] raise a concern about drawing generalized conclusions about human psychology and behavior from experiments on “Western, Educated, Industrialized, Rich and Democratic (WEIRD) societies”. Their concern is reflected in several experiments which were conducted in "small-scale" societies in [8]. More specifically to our study, in research on perspective-taking among individuals from collectivist versus individualist societies, Wu and Keysar [9] and Wu et al. [10] found that for tasks in which they needed to consider the viewpoint of another person, subjects from China performed better and self-corrected more rapidly. In our asymmetric matching pennies games, considering another person’s perspective corresponds to paying attention to the other player’s payoffs, which is exactly what is needed for subjects to reach equilibrium strategies. Subjects are unable to infer the mixed-strategy equilibrium in these games by merely focusing on their own payoffs.

Our paper contributes to the behavioral game theory literature which seeks to understand the discrepancies between theoretical mixed-strategy Nash equilibria and their implementation by human subjects in the laboratory. Ochs [3] finds violations of the own-payoff invariance predicted by mixed-strategy equilibria in a series of matching pennies games. McKelvey and Palfrey [11] propose Quantal Response Equilibrium (QRE), or stochastic best response, as a better fit to the data from several experimental studies, including [3]. Continuing in the approach of employing QRE in explaining own-payoff effects, Goeree et al. [5] propose risk-averse players in a QRE setting in order to accommodate the tendency of QRE to overpredict own-payoff effects. McKelvey et al. [4] conduct experiments on games similar to those in [3] and fit a QRE model with player heterogeneity to explain individual-level own-payoff effects. Martin et al. [12] find that chimpanzees play closer to equilibrium in asymmetric matching pennies games than humans, and they suggest an evolutionary interpretation of game theory. More recently, Friedman and Zhao [13] examine factors affecting the frequency of the played strategy in asymmetric matching pennies games. They find subjects’ behavior depends on the matching method, time protocol and whether subjects can explicitly specify the mixed strategy.

Our study differs from these works primarily in terms of the results obtained, which are substantially closer to the equilibrium prediction. Notably, our Row players did not show the strong own-payoff tendencies found in other papers. On the other hand, while our Column players did not play as closely to their equilibrium strategy as found in [2], they moved significantly towards equilibrium over time, suggestive of learning dynamics. In a Quantal Response Equilibrium framework with the possibility of other-regarding preferences, the behavior in our experiments corresponded the best to a specification with a positive spite parameter [6,14]. While the equilibrium prediction holds up relatively well in the aggregate, we also explore individual differences and possible explanatory factors. Randomizing close to one’s equilibrium strategy is significantly predicted by previous formal training in game theory for Row players, although not for Column players.

Other relevant papers have proposed different explanations for several of the Ten Intuitive Contradictions in [2]. Erlei [15] proposes that social preferences can explain most of the ten contradictory findings, including the asymmetric matching pennies puzzle. Eichberger and Kelsey [16] argue that ambiguity aversion over the other player’s strategy choice can explain [2]’s contradictions for normal form games, including the matching pennies case.

Finally, our paper is related to the larger literature on behavior in games with mixed-strategy equilibria, originating with [17,18]. Shachat [19] investigates subjects’ failure to play the unique minimax mixed-strategy equilibrium and derives a method to distinguish between subjects’ mixed-strategy intentions and their actual implementations. Shachat et al. [20] develop a statistical model to detect mixed-strategy play in repeated games. Belot et al. [21] find that in matching pennies games, subjects who can observe their opponents gestures have a tendency to automatically imitate their opponents’ action. Brocas and Carrillo [22] had children and adolescents play a hide-and-seek game and found that most participants favor the high-value location, even though only the seeker in theory can maximize payoffs in this way Külpmann and Kuzmics [23] propose a new method to compare theories under mixed-strategy predictions. They find that although Nash equilibrium with risk aversion is among the best predictors, it cannot correctly predict the behavior of the own-payoff-seeking player in asymmetric matching pennies and rock–scissors–paper games.

The remainder of this paper proceeds as follows: Section 2 describes our experimental setup; Section 3 provides our aggregate results; Section 4 explores time dynamics; Section 5 discusses individual heterogeneity; Section 6 accounts for the results using ϵ-equilibrium and Quantal Response Equilibrium concepts, as well as additional results from fixed partner matching treatments, which yielded similar results; and Section 7 concludes and discusses.

## 2. Experiment Description

We used the asymmetric matching pennies games proposed by Goeree and Holt [2]. Their “Asymmetric Matching Pennies” is identical to our “Higher-type Asymmetry” game, and their “Reversed Asymmetry” is identical to our “Lower-type Asymmetry” game, where “reverse” in their terminology refers to the pure strategy B having the highest possible payoff for Row player rather than T, as in the original asymmetric games. Figure 1 shows the two games and their equilibrium strategy profiles, which as usual are found by each player randomizing among their pure strategies so as to make the other player indifferent between either of their own pure strategies.

Figure 2 shows the results from Goeree and Holt [2], which were achieved on 50 subjects in 5 cohorts of 10, randomly matched for one-shot play. Payoffs in the table are expressed in cents. Their results show a striking violation of the equilibrium prediction. Compared with the symmetric matching pennies case, where subjects randomized in a manner very close to that predicted by the equilibrium (T: 0.5, B: 0.5; L: 0.5, R: 0.5); the other two games show that the Row player seems to be highly driven towards the pure strategy which has the potential of yielding their highest payoff opportunity. The violation is stark in that 96% of the Row players chose Top in the asymmetric case and 92% of subjects chose Bottom in the reversed case. The Column players tended to correctly anticipate Row players’ behavior, best-responding accordingly. Row players in similar higher-type asymmetric matching pennies games tested in [3,4] tended to deviate from the mixed-strategy Nash equilibrium in favor of the potential highest payoff action by magnitudes of around 10 to 15 percentage points, compared with the 40 percentage point deviation found in the one-shot Goeree and Holt experiment [2]. (Note for example, in the equivalent of the Higher-type Asymmetry game in [3], the Row player is estimated to play their potentially highest payoff yielding action about 63% of the time, compared with the 50% predicted by Nash equilibrium. In [4], the analogous proportions for variations in the Higher-type Asymmetry game are between 59% and 64%. Goeree et al. [5] test the fit of a Quantal Response Equilibrium model with risk aversion on the data of [4], as well as conduct experiments on a modified version of the games, specifically designed to test the role of risk aversion).

Our experimental sessions were conducted on 10 June 2012 in the Tsinghua University School of Economics and Management, Economic Science and Policy Experimental Laboratory (ESPEL). Each session lasted about one hour and consisted of two treatments. Subjects were recruited using the ORSEE system. The experiments were conducted using z-Tree software [24].

Table 1 shows the summary of each session in terms of treatment ordering, partner matching, subject earnings and number of subjects. In each session, subjects were paired randomly each period to play the game. Each treatment within each session contained 50 rounds of asymmetric matching pennies, with a history table which listed subjects’ personal history of play and winnings in each period. Subjects were randomly assigned their roles (Row or Column player) and maintained fixed roles within each treatment of their session. The game was described to subjects using sentences describing each player’s payoff conditional on their choice and their partner’s choice, rather than teaching subjects how to read a game matrix. The details of the experimental instructions can be found in the Appendix A.

Compared with other studies in the literature, our experiment uses the exact “Asymmetric matching pennies” and “Reversed asymmetry” games tested in [2]. However, our design differs from [2] in that our subjects knowingly play each game for 50 rounds. Our experiment shares this repetition feature in common with [4]. An additional shared feature with [2,4] is that we measure strategies by asking subjects to choose actions in each round, rather than attempting to elicit mixed strategies directly. Ochs [3], on the other hand, asks subjects to declare the number of each possible action to be taken out of the ten upcoming rounds. Our design, Ochs [3] and McKelvey et al. [4] all provide the full history of actions and payoffs earned for games participated in to each player.

Overall, our experimental design is quite similar to the experiments conducted in [4].

## 3. Aggregate Results

Table 2 shows the frequency of pure strategies played in the aggregate, for each session, expressed for brevity in terms of the frequency of B choices by Row players and the frequency of R choices by Column players. The randomization as predicted by equilibrium is shown in the leftmost column.

Our results in both Sessions 1 and 2 are quite different from the results in [2] and other studies. Our Row players tended to play very close to the equilibrium prediction on average—even more so in the case of the Higher-type Asymmetry compared with the Lower-type Asymmetry game. Our results for Row players in the Higher-type Asymmetry game are three percentage points away from the equilibrium on average, shown to be indistinguishable from the Nash equilibrium mixture. This pattern contrasts with the 10 to 15 percentage point deviations from equilibrium found in [2,3,4,5]. In our Lower-type Asymmetry treatments, Row players adhered slightly less to the equilibrium proportions than our Higher-type Asymmetry results but were still far from the level of deviation observed in Goeree and Holt’s reversed asymmetry results [2]. A one-sample *t*-test of Nash equilibrium play as the null hypothesis, using individual player frequencies of play over the 50 rounds as the unit of observation, shows that Row players in the Higher-type Asymmetry game were statistically indistinguishable from Nash equilibrium (*t*-statistic: −0.9822, *p*-value: 0.3317). Row players in the Lower-type Asymmetry game played differently from the Nash equilibrium at the 5% level (*t*-statistic: 2.4921, *p*-value: 0.0168). Column players in both games played statistically differently than the Nash prediction in the aggregate (Higher-type Asymmetry: *t*-statistic: −6.3193, *p*-value: 0.0000; Lower-type Asymmetry: *t*-statistic: 8.5884, *p*-value: 0.0000).

Our results suggest that Row players in asymmetric matching pennies games do not merely respond to their own most attractive payoff possibilities, as suggested in the literature. Compared with the experiment in Goeree and Holt [2], our design asked subjects to play each game several times, while reminding them of their history of play in each round, similarly to [3,4,5]. These features of the experiment are likely to have at least partially assisted Row players in randomizing according to their equilibrium strategy.

In addition to differences in the magnitude of deviation, a key distinction between our results and the findings in [3,4] is that own-payoff effects are persistent and significant in their data (in the case of [4]), at the individual level), while our results do not bear this pattern. That is, when the asymmetry in a player’s payoffs across actions is greater, own-payoff effects suggest more play of the associated high-payoff action. As shown in Table 2, this pattern is at least qualitatively reversed among our subjects, with slightly stronger average own-payoff tendencies among Row players occurring in the Lower-type Asymmetry game than in the Higher-type Asymmetry game. The data are too noisy, however, to statistically reject or confirm the own-payoff effect pattern.

Our Column players generally adhered slightly less well to the equilibrium prediction compared with Goeree and Holt’s results [2] and in comparison with our own Row players. In the Higher-type Asymmetry treatment, Column players tended to underplay R, while in the Lower-type Asymmetry treatment, Column players tended to overplay R. However, as the next section shows, Column players tended to converge towards equilibrium frequencies over time. For Row players, the assisting feature seemed to be the prospect of repetition itself rather than learning dynamics.

## 4. Time Dynamics

So far, we have only discussed the aggregate proportions of pure strategies played in the treatments. An additional question is whether subjects’ initial responses adhered to the equilibrium frequencies. Previous experiments on asymmetric matching pennies games often find learning behavior over periods of play, but there are experiments on other games which show that the time horizon has little effect [25].

If initial responses adhered well to equilibrium frequencies, it suggests that it is not the implementation of repetition which induces Row players to equilibrium but the prospect of playing the game several times which encourages players to play in a randomized manner. One possibility in the one-shot game is that players know that they should be indifferent between their two pure strategies; yet, when actually implementing that randomization, they prefer the action which has the possibility of yielding the high payoff as a tie-breaker.

Table 3 shows the initial responses of subjects in the first round, as well as during the first two rounds of play. While the top panel shows some preference for the action with one’s own highest payoff, once the first two rounds of play had occurred, Row players were playing much closer to equilibrium. Column players in the case of the second treatments of the sessions (nonbold) were closer to equilibrium proportions than in the aggregate over all rounds but were farther from equilibrium frequencies in the cases of the first treatments of the session (bold).

The initial responses suggest that while playing many rounds is not necessary for frequencies of actions close to the equilibrium prediction, subjects’ knowledge of the fact that they will play several rounds may help in realizing equilibrium frequencies. However, one question is whether over time the frequencies move even closer to equilibrium. We note that in our aggregate results, Row players already play quite close to equilibrium frequencies and seem to reach close to such frequencies after just the initial rounds of play. On the other hand, Column players have room to improve over time. We estimate a simple linear regression to detect the time trend in subjects’ play, with the proportion of the specified pure strategy played at time t as the dependent variable. A constant and a linear time trend are the explanatory variables.
$$\begin{array}{c} proportion_{t}=\alpha +\beta \times t+\epsilon_{t} \end{array}$$

As in the previous section, we are primarily interested in picking up aggregate trends in the pure strategies played. Thus, we estimate only aggregate coefficients while assuming that the errors are uncorrelated. We examine individual player heterogeneity in Section 5.

As Table 4 shows, in three out of the four treatments, Column players’ adjustment of pure strategies over time was significant in the correct direction towards the equilibrium mixed strategy. For example, in the Session 1 Higher-type Asymmetry treatment, Column players played R 71% of the time, whereas equilibrium would suggest they play R 87.5% of the time. The regression shows that Column players played R about 66% of the time as a baseline and increased their percentage of R play 0.25% points in each round (or an increase of 1% point every 4 rounds) on average. Similarly, in the Lower-type Asymmetry game, Column players were expected to play R 9% of the time, yet on the whole, they played R about 3 times as often as this equilibrium prediction. However, as the time trend coefficients show, Column players adjusted their play of R significantly downward over time. Row players tended to show no particular time adjustment in their decisions.

The exception to this pattern is the Higher-type Asymmetry game in Session 2, where Row players adjusted their strategies in the correct direction but Column players showed no significant adjustment towards equilibrium.

## 5. Individual Analysis

While our results so far have been presented in the aggregate, one question is how well individual subjects adhere to the equilibrium. We are interested in the individual proportions of B and R played for the Higher-type Asymmetry game and the Lower-type Asymmetry game. Our experimental results seem to suggest that most of the ability to fit the equilibrium prediction is from the cross-sectional dimension rather than within subjects.

Each panel in Figure 3 shows the proportion of the specified pure strategy (B for Row player and R for Column player) on the x-axis and the number of subjects on the y-axis. The equilibrium strategy proportion is denoted with the red vertical line. As the top row of panels shows, there was wide individual variance in the proportion of B played by Row players across the entire 50 rounds. There is a substantial amount of heterogeneity, even though when averaged across individuals, the proportion is close to equilibrium. The bottom row of panels shows the analogous charts for Column players’ play of R. The results are supportive of equilibrium in that almost all individuals were randomizing in the correct half of the possible range; yet, most subjects were not implementing equilibrium on an individual basis.

Figure 4 shows essentially the same story for the Lower-type Asymmetry game case. Note that Column players’ distribution of personal randomization is skewed in the correct direction towards the equilibrium prediction in both Figure 3 and Figure 4, even though the equilibrium probabilities of playing R are nearly opposite to one another. That is to say, the individual frequencies of play were in the right direction, with virtually no subjects playing R less than 50% of the time when they should be playing R 87% of the time and virtually no subjects playing R more than 50% of the time when they should be playing R about 9% of the time.

We also conducted a postexperiment survey in which we asked subjects a short series of questions about their personal background characteristics, as well as questions about whether they cared about their partners’ payoffs and their experience with the field of game theory (Note that our subjects reported a relatively high rate of personal exposure (either formal or informal) to game theory. Some experiments (notably [4] in the literature reviewed herein) specifically recruit subjects with limited or no prior training in game theory. The benefits of this approach are clear in that it avoids a situation where the experiment merely tests subjects’ prior learned knowledge. However, there can also be drawbacks to this prescreening approach, particularly if the population that subjects are drawn from is generally inclined to be interested in strategic and game theoretic issues. In particular, the study of strategic interaction has been a traditional Chinese intellectual activity for thousands of years (See, for example, the renowned 2500-year-old classical Chinese text: *The Art of War* by SUN Wu). In the conclusion, we discuss the potential appeal of game theory in Chinese society, and leave further exploration of this connection to future work).

In terms of prior experience with game theory, we asked them three questions: Have you ever heard of game theory? Have you taken a course on game theory? Do you have any books or academic articles on game theory? Remarkably, all of our subjects claimed to have heard of game theory before; we omit this variable from the analysis since it has no explanatory power in behavior in our experiment. A total of 32% of our subjects had taken a course in game theory before. In total, 64% of subjects claimed to have some reading materials on the topic. (Note that we cannot verify whether those students reporting ownership of game theory reading materials had a correct impression of what "game theory" is, a possible reason why we found no significant effect of this variable).

Table 5 and Table 6 show linear regressions, with the individual deviation from the equilibrium mixture, averaged across the large and Lower-type Asymmetry games, as the dependent variable. The results are similar when considering the higher-type and Lower-type Asymmetry games separately. Recall that in our experimental setup, subjects maintained a single role (Row or Column) for both the higher-type and Lower-type Asymmetry treatments.

Table 5, which contains the results for Row players, shows that having taken a game theory course before is significantly (at the 10% level) associated with implementing pure strategy frequencies closer to the theoretical mixture. We do not find a significant effect of any of the explanatory variables for Column players, as seen from Table 6. This may be because Column players experienced significant learning over time (see Table 4) and may be better explained by time dynamics than individual characteristics.

## 6. Accounting for Deviations from Equilibrium Play

Although our results do appear much more adherent to equilibrium play than those previously found in the literature, a natural next question is how well models of noisy equilibrium play can fit our data compared with previous results. To explore this, we consider two possible noisy equilibrium models which have been widely adapted in the literature: ϵ-equilibrium [26] and Quantal Response Equilibrium [11]. In our analysis, we closely follow the approach of [6].

### 6.1. ϵ-Equilibrium

ϵ-equilibrium represents the payoff noise penalty ϵ needed to account for the difference between a particular role’s empirical expected payoff, as determined by the actual frequencies of play, and the payoff from the optimal action given those empirical frequencies. Calculation of the ϵ-equilibrium provides a lower bound on ϵ, from which we can infer the range of behavior that would be consistent with such a minimum penalty.

Figure 5 shows the implied range of behavior (in yellow) under the ϵ-equilibrium assumption for our results (top row) and [2] (bottom row). The Nash equilibrium frequencies for Row player (horizontal axis) and Column player (vertical axis) and our empirical frequencies are labeled. In each panel, the upper area of yellow corresponds to the Lower-type Asymmetry game and the lower area of yellow corresponds to the Higher-type Asymmetry game. We consider two possible specifications of ϵ, a common ϵ for both Row and Column players (left panels) and allowing Row and Column players to have different values of ϵ (right panels). The common ϵ assumption is used in [6] and is based on the larger of the two ϵ values between Row and Column players. The role-specific ϵ is motivated by the large payoff asymmetry between Row and Column players.

In each panel, the x-axis shows the average randomization of Row players, and the y-axis shows the average randomization of Column players. By comparing the top panels (our results) and bottom panels (Goeree and Holt’s results [2]) in Figure 5 in terms of the horizontal and vertical distance between Nash equilibrium and lab results, we can again see the key difference between our results and [2]; our Row players are much closer in their randomization to equilibrium than theirs, while our Column players deviated from equilibrium more than theirs. This is also reflected in the range of randomizations, which can be explained by the ϵ-equilibrium. As the upper right panel shows, the ϵ-equilibrium with role-specific ϵ allows pinpointing of our empirical result quite precisely. The same degree of precision is not obtained in the case of the common ϵ assumption (top left panel), due to the fact that the Column player’s ϵ is quite high (Note that $\epsilon_{Row}$ and $\epsilon_{Column}$ can be found as follows using the payoff structure and empirical strategy frequencies: Higher-type Asymmetry game: (32*0.2657 + 4*0.7343)*0.53045 + (4*0.2657 + 8*0.7343)*0.44955 + $\epsilon_{Row}$ ≥ 32*0.2657 + 4*0.7343; $\epsilon_{Row}$ ≥ 2.1145; (4*0.53045 + 8*0.46955)*0.2657 + (8*0.53045 + 4*0.46955)*0.7343 + $\epsilon_{Column}$ ≥ 8*0.53045 + 4*0.46955; $\epsilon_{Column}$ ≥ 0.0647. Lower-type Asymmetry game: (4.4*0.72095 + 4*0.27905)*0.4029 + (4*0.72095 + 8*0.27905)*0.5971 + $\epsilon_{Row}$ ≥ 4*0.72095 + 8*0.27905; $\epsilon_{Row}$ ≥ 0.3335; (4*0.4029 + 8*0.5971)*0.72095 + (8*0.4029 + 4*0.5971)*0.27905 + $\epsilon_{Column}$ ≥ 4*0.4029 + 8*0.5971; $\epsilon_{Column}$ ≥ 0.2168. The common ϵ specification merely takes the larger of the two roles’ϵ).

### 6.2. Quantal Response Equilibrium

The notion of Quantal Response Equilibrium was proposed by [11]. Related extensions using experimental data from matching pennies games include [6,27,28,29]. In this study, we use the same method as [6], which shows that Goeree and Holt’s results [2] can be accounted for in a QRE framework with altruistic other-regarding preferences. However, the behavior in our study is better accounted for with a positive spite parameter, instead of altruistic preferences towards other players. In allowing for spite, we incorporate in each player’s total payoff, a payoff subtraction of the other player’s payoff, weighted by parameter α [14].

This is also reflected in the range of randomizations, which can be explained by the ϵ-equilibrium. As the upper right panel shows, the ϵ-equilibrium with role-specific ϵ allows pinpointing of our empirical result quite precisely. The same degree of precision is not obtained in the case of the common ϵ assumption (top left panel), due to the fact that the Column player’s ϵ is quite high. (Note that Figure 6 and Figure 7 show the randomizations which can be accounted for in a QRE framework with the range of quantal response parameters, λ∈[0,∞), with the illustrated points radiating out from the center (fully random) corresponding to increasing values of λ. The QRE model converges to the Nash equilibrium randomization as λ→∞. However, as the figures show, variation in the QRE parameter alone may not fully explain our results. A number of previous experiments have shown that behavior can be influenced by social preference [30,31,32,33,34,35,36]. We have successfully found the appropriate social preference parameter α from the framework of [14] which can match the data. The point “New Nash equilibrium” in the Figures corresponds to the Nash equilibrium under the specified social preference parameter.

In the Lower-type Asymmetry game, our subjects (see upper section of Figure) can be explained by QRE with spite parameter 0.2, as illustrated in Figure 6. We note the contrast with the Goeree and Holt (GH) results [2], which conversely requires positive altruism in this framework to explain the empirical result, noticing that the GH randomizations lie on the opposite side of the original Nash equilibrium QRE path compared with our results.

Our subjects in the Higher-type Asymmetry game (see lower section of diagram) require a much higher spite parameter to explain the result. Figure 7 shows QRE with and without the spite parameter value 1.2. Our subjects’ behavior in the Higher-type Asymmetry game is nearly explained by this specification. Once again, the GH result is on the opposite side of the original Nash equilibrium QRE path compared with ours, requiring positive altruism to explain it.

These QRE results highlight a key apparent difference between our subjects and the subjects from the US in the previous experiments: The behavior of the US-based subjects in previous experiments generally corresponds to altruistic preferences over payoffs, while the behavior of our subjects seems to be better explained by spiteful motives. That is, our subjects behave as though they were receiving negative utility from the other player’s gains. We also note that the magnitude of spite needed to explain the result is much higher in the Higher-type Asymmetry game (α=1.2) than the Lower-type Asymmetry game (α=0.2), implying that the greater the payoff possibility, the greater the spite. This suggests that at least within this framework, players’ social preferences may be context-dependent; when facing a larger payoff asymmetry in a competitive environment, players may adopt more heavily spiteful social preferences. Further work is needed to determine the robustness of this effect.

### 6.3. Fixed Partner Matching

We also conducted sessions with fixed partner matching over the stage game repetitions. Although most of the literature on experimental asymmetric matching pennies implements random partner matching, due to the unique (mixed strategy) equilibrium of the stage game, in our finitely repeated setting, the classical theory makes the same prediction regardless. It is also interesting to check whether subjects’ behavior varied when they were dealing with just one opponent throughout.

Consequences of learning in games with unique mixed-strategy equilibrium have been analyzed in [37], and strategy dynamics have been analyzed empirically in [38,39]. These studies suggest that repeated interactions with a specific opponent could yield different results for behavioral reasons. As in the previous treatments, players’ roles were kept fixed throughout each session. Table 7 provides the session summary statistics for the fixed partner matching treatments.

The aggregate statistics in Table 8 indicate that for Row players, play was close to equilibrium and, on average, even closer to equilibrium than in the random rematching treatments. The results for Column players tended to deviate a bit further from equilibrium frequencies compared with the random rematching case. As in the random rematching treatments, "B" was slightly more popular than "T" among Row players in the Lower-type Asymmetry game but still not near the level of popularity found by [2]. The one-sample *t*-test of Nash equilibrium as the null hypothesis, using individual player frequencies of play over the 50 rounds as the unit of observation, shows that Row players in the Lower-type Asymmetry game were statistically indistinguishable at the 5% level from Nash equilibrium (*t*-statistic: 1.5437, *p*-value: 0.1307). Row players in the Higher-type Asymmetry game played differently from the Nash equilibrium at the 5% level, underplaying the high-payoff action (*t*-statistic: 2.1962, *p*-value: 0.0341). Column players in both games played statistically differently than the Nash prediction in the aggregate (Higher-type Asymmetry: *t*-statistic: −12.2472, *p*-value: 0.0000; Lower-type Asymmetry: *t*-statistic: 12.8014, *p*-value: 0.0000).

Table 9 shows that a similar pattern existed when considering just the initial rounds of play, with the mixed-strategy equilibrium frequency being exactly reached in the first round for three out of the four fixed partner treatments.

Figure 8 and Figure 9 show the individual-level results for the Higher-type Asymmetry and Lower-type Asymmetry games, respectively. Compared with the random rematching treatments, the individual frequencies of play tend to display less variance, particularly for the Row players, as seen from the dispersion in frequencies in the top panels of Figure 8 compared with the top panels of Figure 3 and the top panels of Figure 9 compared with the top panels of Figure 4. For the Column players, compared with the case of random rematching, subjects in the fixed matching treatments tended to center their mixing frequencies farther from the equilibrium prediction. Overall, it appears that fixed partner matching tends to draw play closer to the 50% mark for both player roles compared with the case of random rematching. (This makes intuitive sense for Row players, supposing a Row player in the random rematching treatment thinks of themself as one of many possible “types” in the population whose aggregated average frequencies of play correspond to the Nash equilibrium. They may then find it reasonable to play B with 10% likelihood, so long as others play with other likelihoods to average across Row players as 50%. In the case of fixed partner matching, this behavioral argument does not hold, which may draw players closer to the 50% equilibrium play. The same pattern does not seem to hold for Column players, however, who tend to deviate more from equilibrium play in the fixed partner matching treatments).

## 7. Conclusions

In this study, we conducted experiments on the asymmetric matching pennies contradiction posed by [2], which has been robustly confirmed in the literature as being driven by “own-payoff effects” [3,4,5]. In our experiments, not only did we find randomization frequencies closer to Nash equilibrium strategies than in previous studies (particularly for the Row player with the high payoff potential), but we also did not find the signature pattern of own-payoff effects, where the higher the associated potential payoff, the more a player gravitates to the associated action.

When we consider the concept of ϵ-equilibrium, particularly in the case of role-specific ϵ, our results are more precisely pinpointed on the Row players’ dimension within the ϵ-equilibrium framework than previous experimental results. This is due to the fact that our Row players adhered quite closely to equilibrium frequencies, while Column players deviated significantly.

In the Quantal Response Equilibrium framework allowing for altruism or spite, this corresponded to a small positive spite parameter to match our Lower-type Asymmetry results, as wll as a large positive spite parameter to match our Higher-type Asymmetry results. This result is in distinct contrast to the results of previous experiments conducted in the US. When analyzed within the Quantal Response Equilibrium framework with possible altruism or spite, the deviations in the previous study were in the direction of altruism [6]. In other words, our subjects behaved in a manner consistent with receiving negative utility from the other player’s positive payoffs. Social preference has been shown to depend on a number of factors, such as culture [40], social identity [41] and religion [42]. The subjects in our experiment seem to display different preferences towards the payoff of their matched partner.

Cultural psychology studies suggest that collectivist cultures foster the ability to take other peoples’ perspective into account more than individualist cultures do. For example, [9] found that Chinese subjects performed better than US subjects in a task involving communicating about placing objects in which they needed to consider a partner’s perspective. In a follow-up study, Wu et al. [10] found that differences in the tendency to take the perspective of others across the two cultures could be attributed to a more rapid and effective correction of egocentric thinking among Chinese subjects. If these tendencies in perspective-taking carry over to non-cooperative game settings, Chinese subjects might be expected to adopt mixed-strategy equilibrium reasoning more readily than US subjects tested in previous studies.

Indeed, our subjects did not display the contradiction of “own-payoff” effects in this unique mixed-strategy equilibrium setting. In particular, our Row players did not overgravitate to the high-payoff strategy compared with the equilibrium prediction on average, especially in the Higher-type Asymmetry game. Our Column players, while not adhering as precisely to the equilibrium prediction on average compared with Row players in our study, showed significant adjustment towards the direction of equilibrium over time. Due to the prevalence of results in the literature which support the matching pennies contradiction, which include both repetition and a visible history of play as we do [3,4,5], it is difficult to infer that any feature our experimental design is responsible for this difference in results. Further work is needed to determine whether cultural differences in imagining the viewpoint of others is responsible for the discrepancy.

An important factor which could contribute in generating these results is the competitive nature of modern Chinese society [43,44,45]. It is widely known that students in China face intense academic competition from a young age at each stage of education. Students have been conditioned to compete with their peers for a high score. This may help to eliminate phenomena such as altruistic preferences over opponents’ payoffs or desires to take trial-and-error-based approaches. This mentality may have led our subjects to take the game and its payoff consequences more seriously than the subjects in the US. It may also be a reason for the prevalent interest in game theory, as self-reported by our subjects.

We can see several directions for future research. First, while the differences between Chinese subjects’ performance in the asymmetric matching pennies games and that of US subjects is striking, much more work is needed to test the hypothesis that Chinese subjects pay greater attention to their opponents’ incentives than Western subjects in non-cooperative games. In the context of behavioral "anomalies" in game theory, some of the findings other than the asymmetric matching pennies contradiction may be robust to Chinese populations. Second, we have only tested the game on Chinese subjects and not on subjects from collectivist but non-Chinese societies. More work is needed to test the robustness of our findings across other collectivist cultures. Finally, we believe that further work using other methodologies can be conducted on these games in China to more precisely pinpoint perspective-taking hypotheses. Methodologies involving biological indicators or search tracking may be one such direction which can help to confirm and better understand a possible cultural-based explanation for the differences between our findings and those in the prior literature.

## Figures and Tables

**Figure 1 behavsci-13-00757-f001:**
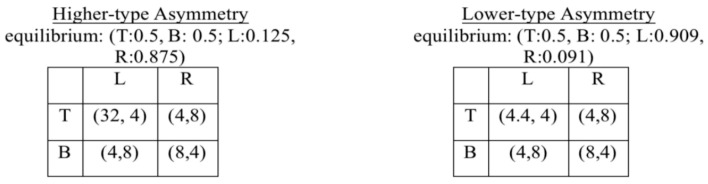
Matching pennies games: Payoffs expressed in Experimental Currency Units (ECU), which convert to CNY at the rate 10 ECU = 1 CNY.

**Figure 2 behavsci-13-00757-f002:**
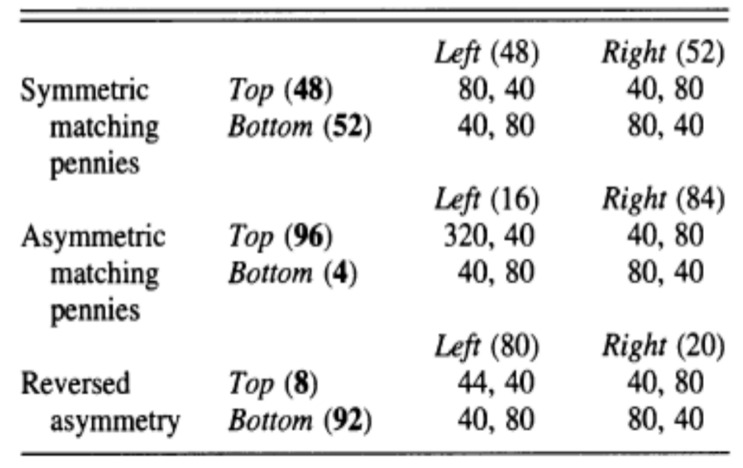
Goeree and Holt results [2].

**Figure 3 behavsci-13-00757-f003:**
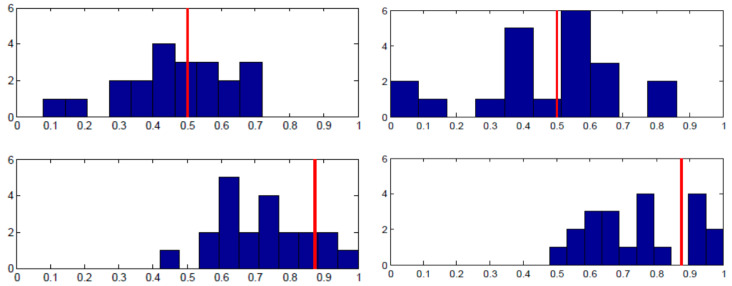
Higher-type asymmetry individual results. **Left panels** —Session 1; **Right panels**—Session 2; **Top panels**—Row player, B frequency on horizontal axis; **Bottom panels**—Column player, R frequency on horizontal axis; vertical red line: equilibrium strategy.

**Figure 4 behavsci-13-00757-f004:**
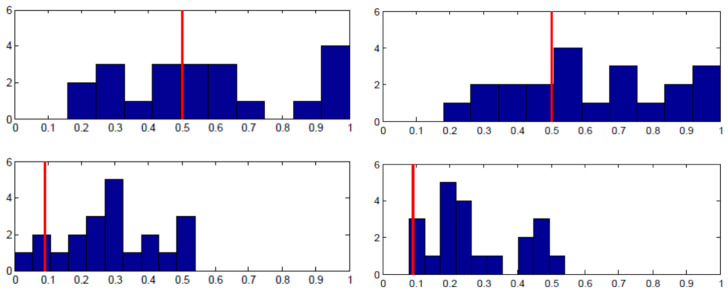
Lower -type asymmetry individual results. **Left panels** —Session 1; **Right panels**—Session 2; **Top panels**—Row player, B frequency on horizontal axis; **Bottom panels**—Column player, R frequency on horizontal axis; vertical red line: equilibrium strategy.

**Figure 5 behavsci-13-00757-f005:**
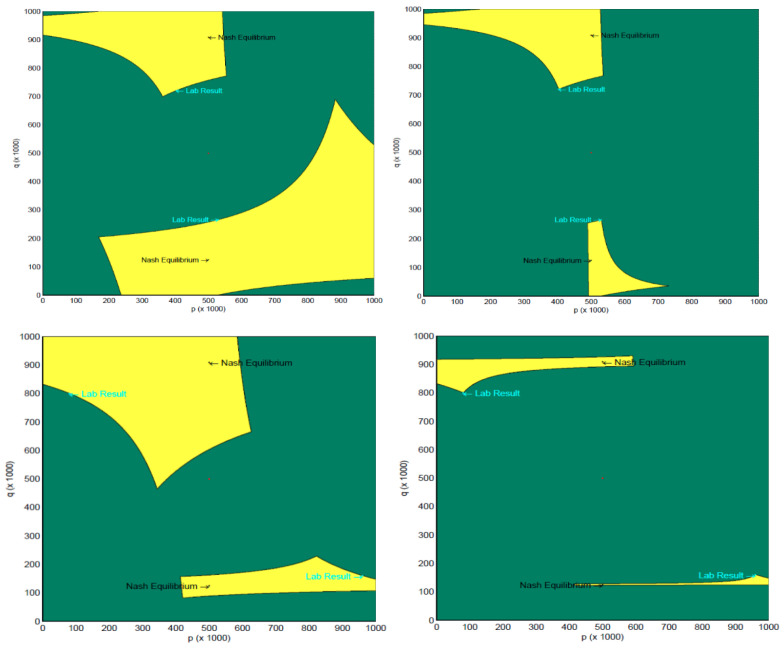
Range of Row (horizontal axis) and Column (vertical axis) players’ randomization explained by ϵ-equilibrium. **Top left**: Our data, single epsilon for Row and Column players; **Top right**: Our data, Row- and Column-specific epsilon; **Bottom left**: Goeree and Holt [2], single epsilon for Row and Column players; **Bottom right**: Goeree and Holt [2], Row- and Column-specific epsilon.

**Figure 6 behavsci-13-00757-f006:**
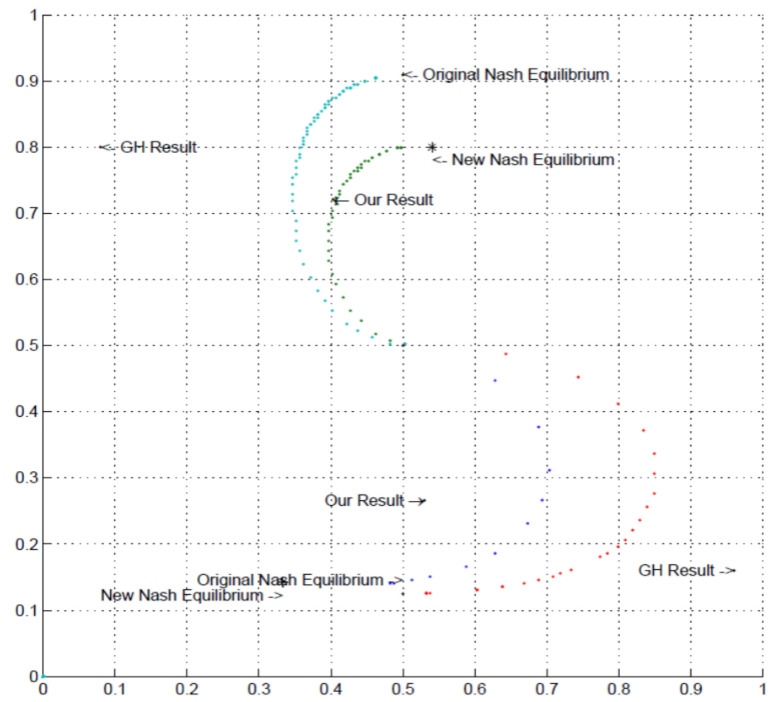
Quantal Response Equilibrium (QRE) randomizations under low spite (α = 0.2) compared with theoretical and empirical results for the Higher-type Asymmetry and Lower-type Asymmetry games (GH denotes Goeree and Holt results [2]; dots radiating from the center correspond to increasing value of QRE λ parameter); * points to New Nash Equilibrium under the specified social preference parameter.

**Figure 7 behavsci-13-00757-f007:**
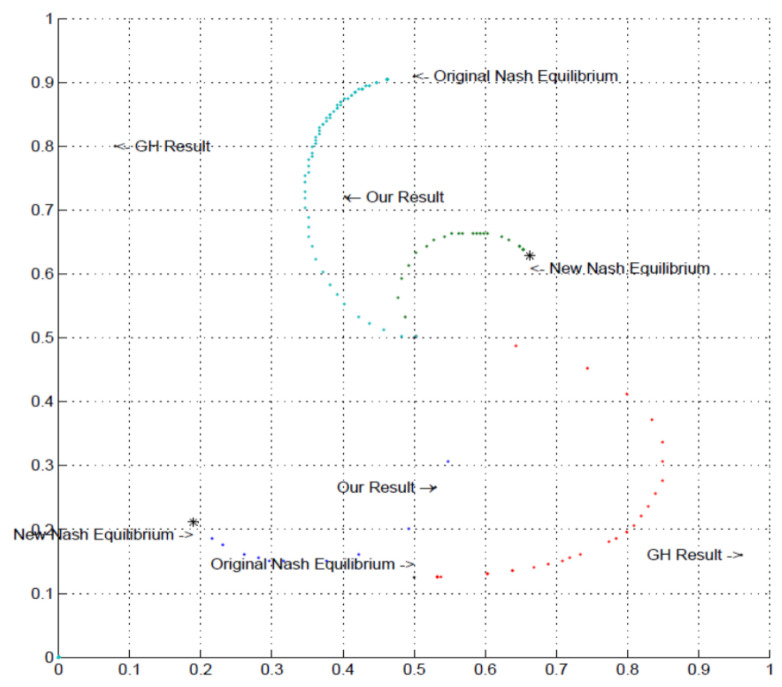
Quantal Response Equilibrium randomizations under high spite (α = 1.2) compared with theoretical and empirical results for the Higher-type Asymmetry and Lower-type Asymmetry games (GH denotes Goeree and Holt results [2]; dots radiating from the center correspond to increasing value of QRE λ parameter); * points to New Nash Equilibrium under the specified social preference parameter.

**Figure 8 behavsci-13-00757-f008:**
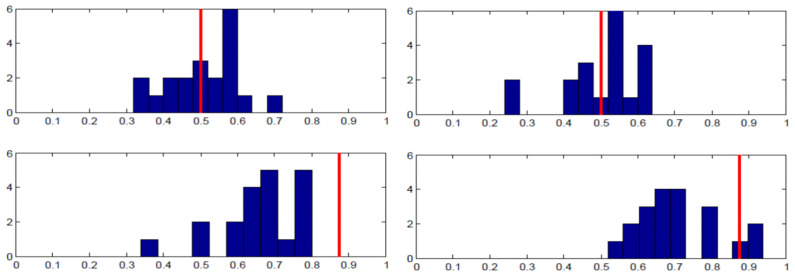
Higher-type Asymmetry individual results (fixed matching). **Left panels**—Session 3; **Right panels**—Session 4; **Top panels**—Row player, B frequency on horizontal axis; **Bottom panels**—Column player, R frequency on horizontal axis; vertical red line: equilibrium strategy.

**Figure 9 behavsci-13-00757-f009:**
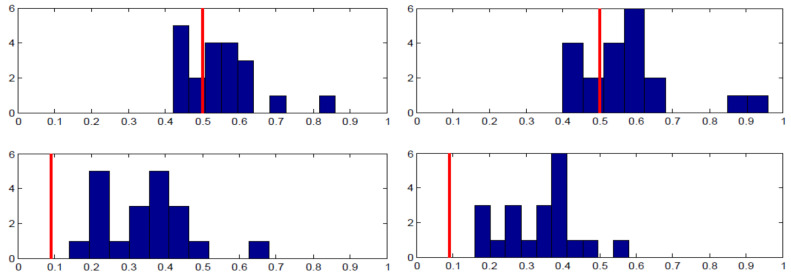
Lower-type Asymmetry individual results (fixed matching). **Left panels**—Session 3; **Right panels**—Session 4; **Top panels**—Row player, B frequency on horizontal axis; **Bottom panels**—Column player, R frequency on horizontal axis; vertical red line: equilibrium strategy.

**Table 1 behavsci-13-00757-t001:** Session Summary.

	Session 1	Session 2
Partner Matching	Random	Random
1st Treatment (Asymmetry)	Higher-type	Lower-type
2nd Treatment (Asymmetry)	Lower-type	Higher-type
Average earnings	75.4	75.3
Standard deviation	8.1	7.7
Minimum earnings	68	67.2
Maximum earnings	103.2	103.2
Number of subjects	42	42

**Table 2 behavsci-13-00757-t002:** Aggregate Statistics.

	Session 1	Session 2
Higher-type Asymmetry (50% B; 87.5% R)	**(47% B; 72% R)**	(47% B; 75% R)
Lower-type Asymmetry (50% B; 9% R)	(58% B; 29% R)	**(62% B; 27% R)**

Note: bold indicates first treatment played.

**Table 3 behavsci-13-00757-t003:** Initial Responses: Pure strategy frequencies in rounds 1 and 2.

	Session 1	Session 2
Higher-type Asymmetry (50% B; 87.5% R)	**First round: (38% B; 71% R)**	(48% B; 90% R)
Lower-type Asymmetry (50% B; 9% R)	First round: (57% B; 33% R)	**(67% B; 38% R)**
Higher-type Asymmetry (50% B; 87.5% R)	**First 2 rounds: (50% B; 71% R)**	(52% B; 90% R)
Lower-type Asymmetry (50% B; 9% R)	First 2 rounds: (52% B; 26% R)	**(60% B; 40% R)**

Note: bold indicates first treatment played.

**Table 4 behavsci-13-00757-t004:** Time Trend Regressions.

	Session 1		Session 2	
Higher-type Asymmetry	row player	column player	row player	column player
Empirical frequency	**51% B**	**71% R**	47% B	75% R
Constant	0.4500 **	0.6588 **	0.5448 **	0.7525 **
Time trend coefficient	0.0007	0.0025 **	−0.0029 **	−0.0003
Lower-type Asymmetry	row player	column player	row player	column player
Empirical frequency	57% B	29% R	**62% B**	**27% R**
Constant	0.5745 **	0.3504 **	0.6083 **	0.3572 **
Time trend coefficient	0.0000	−0.0025 **	0.0004	−0.0033 **

Note: bold indicates first treatment played; ** denotes significance at 95% level.

**Table 5 behavsci-13-00757-t005:** Dependent Variable: Deviation from Equilibrium Mixture (Row players).

	Coefficient	SE
Gender	−0.0348	(0.0446)
Class Year	0.0028	(0.0186)
Game Theory Course	−0.0889 *	(0.0513)
Game Theory materials	0.0481	(0.0526)
Other regarding payoffs	0.0668	(0.0516)
Constant	0.1787	(0.1298)

Note: Standard errors in parentheses. * significant at 10% level.

**Table 6 behavsci-13-00757-t006:** Dependent Variable: Deviation from Equilibrium Mixture (Column players).

	Coefficient	SE
Gender	−0.0151	(0.0323)
Class Year	−0.0100	(0.0122)
Game Theory Course	0.0302	(0.0342)
Game Theory materials	−0.0452	(0.0344)
Other regarding payoffs	0.0159	(0.0452)
Constant	0.2437 **	(0.0916)

Note: Standard errors in parentheses. ** significant at 5% level.

**Table 7 behavsci-13-00757-t007:** Session Summary.

	Session 3	Session 4
Partner Matching	Fixed	Fixed
1st Treatment (Asymmetry)	Higher-type	Lower-type
2nd Treatment (Asymmetry)	Lower-type	Higher-type
Average earnings	76.6	75.6
Standard deviation	11.3	9.3
Minimum earnings	61.2	63.7
Maximum earnings	123.7	98.1
Number of subjects	40	40

**Table 8 behavsci-13-00757-t008:** Aggregate Statistics (Fixed Matching).

	Session 3	Session 4
Higher-type Asymmetry (50% B; 87.5% R)	**(51% B; 66% R)**	(51% B; 71% R)
Lower-type Asymmetry (50% B; 9% R)	(54% B; 35% R)	**(57% B; 34% R)**

Note: bold indicates first treatment played.

**Table 9 behavsci-13-00757-t009:** Initial Responses (Fixed Matching): Pure strategy frequencies in rounds 1 and 2.

	Session 3	Session 4
Higher-type Asymmetry (50% B; 87.5% R)	**First round: (50% B; 60% R)**	(50% B; 75% R)
Lower-type Asymmetry (50% B; 9% R)	First round: (50% B; 30% R)	**(70% B; 30% R)**
Higher-type Asymmetry (50% B; 87.5% R)	**First 2 rounds: (45% B; 58% R)**	(53% B; 75% R)
Lower-type Asymmetry (50% B; 9% R)	First 2 rounds: (52% B; 40% R)	**(73% B; 30% R)**

Note: bold indicates first treatment played.

## Data Availability

The datasets used and analyzed in this study are available from the corresponding author upon reasonable request.

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
