# Peer review of "Revisiting the Asymmetric Matching Pennies Contradiction in China"

_behavsci, 2023, doi:10.3390/bs13090757_

Round 1
Reviewer 1 Report
Referee Report for Behavioural Science manuscript No-2535778
Revisiting the Asymmetric Matching Pennies Contradiction in China
Overview of paper.
The author(s) conducted experiments to test quantal response equilibrium (QRE), i.e., the notion that the probability of choosing a given strategy depends on the “quantal” or payoff associated with the strategy. The author(s) report results from experiments that refute QRE.
Minor points
Lines 31-33. It is not clear how the author(s) reached the conclusion that “mixed strategy equilibrium concept may not be an appropriate description of human strategic behavior”. If it holds for symmetry, then it is valid for human strategic behaviour in that context but not otherwise.
Major points
The experiment in this paper was conducted in 2012 and the literature seems to be dated (notwithstanding 3 tangential references dated 2023). More recent, and arguably more related papers include but is not limited to the references below.
References
Gärtner, M., Östling, R. & Tebbe, S. (2023). Do we all coordinate in the long run?. J Econ Sci Assoc 9, 16–33. https://doi.org/10.1007/s40881-022-00125-z
Brocas, I. and Carrillo, J.D. (2022), The development of randomization and deceptive behavior in mixed strategy games. Quantitative Economics, 13: 825-862. https://doi.org/10.3982/QE1769
Külpmann, Philipp, Kuzmics, Christoph (2022), “Comparing theories of one-shot play out of treatment”, Journal of Economic Theory, 205: 105554
Author Response
Please see the attached reply letter.

Reviewer 2 Report
I am familiar broadly with the topic, but am not an expert at depth. The paper was well organized and had clear exposition. The connection to current literature, discussion of importance, and results were clear and convincing. I was unsure as to why the data discussed in the paper is over eleven years old, which did seem odd. However, since that was my only issue I don't see any reason to cause alarm with the paper overall.
Author Response
Please see our attached reply letter.

Reviewer 3 Report
Title: Revisiting the Asymmetric Matching Pennies Contradiction in
china
ID: behavsci-2535778
The introduction, theoretical framework and conclusion parts of this study, which is based on game theory, are well written. The language used by the researchers and their style of transferring the article are professional. Graphical descriptions are powerful. The only problem I see with the article is that the reference notation in the abstract is not suitable for scientific articles. Authors should revise their abstract. After this minor revision, the article can be accepted for publication.
Author Response
Please see our attached reply letter.
